# Replacing Animal-Based Proteins with Plant-Based Proteins Changes the Composition of a Whole Nordic Diet—A Randomised Clinical Trial in Healthy Finnish Adults

**DOI:** 10.3390/nu12040943

**Published:** 2020-03-28

**Authors:** Essi Päivärinta, Suvi T. Itkonen, Tiina Pellinen, Mikko Lehtovirta, Maijaliisa Erkkola, Anne-Maria Pajari

**Affiliations:** 1Department of Food and Nutrition, University of Helsinki, P.O. Box 66 (Agnes Sjöbergin katu 2), University of Helsinki, 00014 Helsinki, Finland; essi.paivarinta@helsinki.fi (E.P.); suvi.itkonen@helsinki.fi (S.T.I.); tiina.pellinen@helsinki.fi (T.P.); maijaliisa.erkkola@helsinki.fi (M.E.); 2Institute for Molecular Medicine Finland, University of Helsinki, P.O. Box 20, 00014 Helsinki, Finland; mikko.lehtovirta@helsinki.fi

**Keywords:** plant protein, animal protein, whole diet, protein, fibre, lipoproteins, plant-based, dietary intake, dietary fatty acids

## Abstract

Increased consumption of plant-based foods and decreased consumption of animal-based foods is recommended for healthy diets and sustainable food production. We investigated the effects of partial replacement of dietary animal proteins with plant-based ones on intake of energy-yielding nutrients, fibre, and plasma lipoproteins. This 12-week randomised clinical intervention comprised 107 women and 29 men (20–69 years) in three diet groups with different dietary protein compositions (“ANIMAL”: Animal 70%/plant 30%; “50/50”: Animal 50%/plant 50%; “PLANT”: Animal 30%/plant 70%; all: Protein intake 17 E%). Nutrient intakes were assessed by 4-day food records. Saturated fat intake (E%) was lower and polyunsaturated fatty acid intake (E%) higher in the PLANT and 50/50 groups compared to the ANIMAL group (*p* < 0.001 for all). Fibre intake was higher in the PLANT (*p* ˂ 0.001) and 50/50 (*p* = 0.012) groups. Total and LDL cholesterol were lower in the PLANT than in the ANIMAL group (*p* = 0.003 for both) but no differences in HDL cholesterol or triglycerides were observed (*p* > 0.05). Replacing animal protein with plant protein sources in the diet led to an increased fibre intake and improved dietary fat quality as well as blood lipoprotein profile. Flexitarian diets could provide healthy and more sustainable alternatives for the current, predominantly animal-based diets.

## 1. Introduction

Climate change and increased understanding of environmental impacts of food production have resulted in an urgent need to develop sustainable food systems. EAT-Lancet Commission’s report [1] calls for a shift towards more plant-based diets to meet the UN Sustainable Development Goals and the Paris Agreement, and presents the Planetary Health Diet as a reference for healthy and sustainable diet for the growing population. The Planetary Health Diet is a flexitarian diet consisting mainly of plant-based foods and containing only small amounts of animal-based foods. It aims to more than double the consumption of fruits, vegetables, legumes, and nuts, and more than half the global consumption of red meat and added sugars by 2050 [1]. To achieve this goal, a major transformation from current westernised diets to mainly plant-based diets is needed.

Dietary patterns in high-income countries are characterised by a high proportion of animal-based foods. In US adults, 62% of dietary protein originated from animal and 30% from plant protein sources [2]. The proportions of non-dairy animal proteins and dairy proteins were 46% and 16% of total protein intake; the two top food categories were chicken and beef, which provided 26% of non-dairy animal protein and 13% of total dietary protein intake. In Finland, animal-based foods provided altogether 69% of total protein intake in men and 65% in women, whereas the proportions of plant-based foods were 25% and 29%, respectively [3]. The proportion of cereals—the major source of plant protein—was 17% of total protein intake both in men and in women. The average weekly consumption of red and processed meat was 762 g in men and 392 g in women, while Finnish national nutrition recommendations suggest an upper limit of 500 g for weekly consumption [4], World Cancer Research Fund recommends a range of 350–500 g/wk [5], and Planetary Health Diet a range of 0–200 g of red meat per week [1]). Current data from the Nordic Countries show that the typical diets in the area are neither healthy nor environmentally sustainable [6]. Thus, there is a need for a dietary shift towards decreasing especially the consumption of animal-based protein sources [6].

Epidemiological studies have reported varied associations between dietary proteins and different protein sources and risks of chronic diseases, including cardiovascular disease (CVD), type 2 diabetes (T2D), colorectal cancer (CRC), and mortality [7]. Higher plant protein intake was associated with lower all-cause and CVD-related mortality, and replacement of red and processed meat protein with plant protein was associated with lower total, cancer-related, and CVD-related mortality [8,9]. Two recent meta-analyses found an association between total and animal protein intake and higher risk of T2D, while the association between plant protein intake was less clear or U-shaped [10,11]. In a large US cohort study, modelling the substitution of plant protein for animal protein was associated with reduced risk of CRC, and in more detail, the reduction in risk was limited to the substitution of protein from cereals for protein from red meat [12]. However, a meta-analysis involving 21 observational studies did not find an association between dietary protein intake and CRC risk, but a high degree of study heterogeneity may have affected the result [13].

Clinical trials have offered insights into the mechanisms whereby dietary protein sources may mediate their effects on risk of chronic diseases. Meta-analyses of randomised controlled trials found that substitution of plant protein for animal protein decreased low-density lipoprotein (LDL) cholesterol, non-high-density lipoprotein cholesterol, and apolipoprotein B [14,15]. Higher protein intake, particularly from plant-based sources, may also be associated with attenuated inflammatory burden in aging populations [16]. In diabetics, replacing animal protein with plant proteins may help to improve glycaemic control and prevent the onset of risk factors associated with CVD [14,17,18].

Most of the previous clinical intervention studies investigating the effects of plant proteins on human health have been carried out using soy as supplementation or as a part of modified diets and with non-healthy subjects [18,19]. However, plant protein sources are many beyond soy and for both environmental and health reasons, their effects as part of a habitual diet as a whole should be investigated. The aim of the present study was to apply a whole-diet approach to study the effects of partial replacement of dietary animal-based proteins with plant-based ones on food consumption, intake of energy-yielding nutrients, fibre, and on plasma lipoproteins. The study was a 12-week randomised clinical trial with healthy adult volunteers.

## 2. Materials and Methods

The ScenoProt intervention study was a randomised clinical trial with three parallel groups. The intervention period was carried out between December 2016 and June 2017 at the Department of Food and Nutrition, University of Helsinki, as a part of a large, multidisciplinary research project “ScenoProt—Novel protein sources for food security”. The study was registered as a clinical trial in ClinicalTrials.gov database (https://clinicaltrials.gov/, trial number NCT03206827), approved by the Coordinating Ethics Committee of the Hospital District of Helsinki and Uusimaa (1651/2016), and carried out according to the principles of the Declaration of Helsinki. Dietary intake of animal and plant-based proteins, carbohydrates, fibre, and fats were analysed as primary outcome measures, and plasma lipoproteins as secondary outcomes.

### 2.1. Study Participants

The participants were recruited using newspaper advertisements, mailing lists and intranet of the University of Helsinki, and a social media platform (Facebook). Eligibility criteria for participants were being healthy and omnivore, willingness to follow a randomly allocated intervention diet for 12 weeks, age of 20–69 years, and body mass index within the range of 18.5 and 35 kg/m^2^. Exclusion criteria were inflammatory bowel diseases, irritable bowel syndrome, coeliac disease, medication for diabetes or hypercholesterolemia, disorders of the endocrine system or lipid metabolism, liver or renal diseases, cancer within the past five years, regular or recent (within the past three months) use of antibiotics, regular use of nutritional supplements, food allergies, eating disorder, extreme sport, smoking, pregnancy, or lactation. Based on screening by a phone call, all the volunteers who met the criteria provided their written informed consent and were invited to a screening test to analyse plasma total cholesterol and plasma glucose after an overnight fast (10–12 h). Participants with plasma total cholesterol ≤6.5 mmol/L and plasma glucose ≤6.9 mmol/L were invited to attend the intervention period. Background data including the use of antibiotics during the last year, amount of sweat-inducing physical activity, use of nutritional supplements, natural products, hormonal contraception, hormone replacement therapy, and use of any kind of medication were collected through a questionnaire. Data on education were collected through a web-based questionnaire. Participants were asked to quit the use of nutritional supplements and natural products two weeks prior to the beginning of the intervention period. Originally, 145 Caucasian participants (113 women, 32 men) were stratified by gender and age and assigned randomly to three intervention groups for 12 weeks.

### 2.2. Diets

The diets were designed using a whole-diet approach, where the amount of all animal-based protein sources except fish and eggs were partly replaced with plant-based protein sources. Protein intake from all diets was planned to be 17 E% while the proportions of animal and plant-based proteins differed as follows: (1) An animal protein-based diet representing an average Finnish diet and containing 70% of proteins from animal sources (red and white meat, fish, eggs, dairy products) and 30% from plant sources (mainly from bread, other cereals, and potatoes); referred later as ‘ANIMAL’, (2) a diet containing equal amounts (50/50) of animal and plant proteins, and no more than 500 grams of red and processed meat per week; referred in the text as ‘50/50’, and (3) a plant protein-based diet with 30% animal and 70% plant proteins; referred later as ‘PLANT’. In 50/50 and PLANT diets, animal-based protein sources were partly replaced with both new and traditional plant protein sources based on peas, beans, faba beans *(Vicia faba)*, oats, nuts, seeds, and tofu. All diets included the same amount of fish (250 g/wk) and eggs (four eggs, on average 220 g/wk as boiled, fried, or included in foods or pastries) (Table 1). To reduce possible intestinal discomfort caused by plenty of plant protein-based products, the subjects assigned to the PLANT diet followed the 50/50 diet during the first week of the intervention and were then transferred to the PLANT diet.

Participants were provided with most of their protein sources by the study including meat and meat products, poultry products, fish and vegetable patties, ready-made meals, frozen and dried pulses as well as new ready-made plant protein-based products, tofu, nuts, seeds, bread, and cereal. The foods offered by the study supplied all the essential protein sources except dairy products and eggs that the participants were advised to use according to the given instructions. Based on the food records at the end of the study, the controlled food items on the intervention diets supplied on average 80% of the daily energy intake in all groups (ANIMAL 80.2%, 50/50 78.5%, PLANT 80.1%). In addition, the participants were asked to go on using their habitual fats on bread and cooking. They were allowed to consume habitual amounts of foods with low protein content, such as fruit, vegetables, juice, confectionery, and alcoholic beverages.

Energy contents of intervention diets were based on average energy consumption of 8400 kJ (2000 kcal) but the participants were advised to maintain a stable weight and to eat according to their appetite. During their weekly visits at the research centre, the participants obtained the foods for the next week but were also weighed by a nutritionist and were given personalised advice to follow their assigned diet. Since the participants were given both foods to be eaten as such as well as food ingredients, they were guided to implement the diet on food level as shown in Table 1.

### 2.3. Assessment of Dietary Intake

Food consumption and nutrient intakes were assessed using 4-day food records filled in by the participants before and at the end of the intervention period. Additionally, participants were advised to fill in 1-day food records during the intervention period every third week. The participants were given both written and oral instructions on how to accurately fill-in the food records and to estimate portion sizes with household measures and food package labels. Food records were instructed to comprise three weekdays and one weekend day within one week. Completed food records were reviewed by a nutritionist, and any missing information was requested from participants.

At the baseline, the dietary intake data for 118 participants were based on 4-day food records whereas data for 11 participants were based on three recording days and for one participant two days, respectively. At the endpoint, 4-day food records were analysed for 117 participants, 3-day *n* = 16, 2-day *n* = 1, and 1-day *n* = 1, respectively. For the endpoint data, the missing (*n* = 4) food records were replaced by separate food records completed by the participants during the intervention period. One incomplete recording day both at the baseline and at the endpoint was rejected. At the baseline four (5-day *n* = 2, 6-day *n* = 1 and 7-day *n* = 1) and at the endpoint one (6-day *n* = 1) participants kept food records for longer than the instructed four days. The food records of two participants at the baseline were not available.

Food records were processed by nutritionists using AivoDiet software (version 2.2.0.1, Aivo Oy, Turku, Finland), which utilises Fineli (Release 16, National Institute for Health and Welfare, Helsinki, Finland, 2013), the National Food Composition Database maintained by the Finnish Institute for Health and Welfare. New food items were added into the database when necessary. For each individual meal, either a suitable recipe found in the database was used or a new recipe was created when needed. Food consumption and nutrient intakes were calculated as daily mean or median intakes from the total food record period. All foodstuffs were classified into 14 food categories that were based on the classification of Fineli with some modifications, e.g., a category of plant-based dairy-like products was added. All plant protein containing main course dishes were placed to vegetable or vegetable dishes category. The share of the sources for energy and nutrient intakes were calculated for each category.

### 2.4. Sample Collection, Biochemical Analyses, and Anthropometric Measurements

At the screening, baseline and end of the intervention period, venous blood samples for biochemical analyses were collected after a 10–12 h fast. Plasma samples from lithium heparin tubes were analysed for total, LDL, and HDL cholesterol as well as triglycerides by accredited standard methods at Helsinki University Hospital Laboratories in Helsinki, Finland.

At the baseline and the end of the intervention, the participants collected 24-h urine samples (one per time point). Urinary urea concentrations were analysed by a photometric method by Indiko automatic analyser (Thermo Clinical Labsystems Oy, Espoo, Finland) at the Department of Food and Nutrition, University of Helsinki, Finland. Inter coefficients of variation were <4.8% and intra-coefficients of variation <2.8%, respectively. Urinary 24 h excretion of urea was calculated based on the collection times and urinary volumes. Urea excretion was used to calculate nitrogen excretion and then protein intake using the Maroni’s formula [20].
U-protein = U-nitrogen (g/d) + 0.031 ∗ bodyweight (kg) ∗ 6.25.

Bodyweight was measured with a calibrated scale to the nearest 100 g at the baseline and at the end of the intervention. Height was measured with a standard stadiometer at the baseline for the determination of body mass index (BMI). Height and weight were measured with participants wearing light clothing and no shoes.

### 2.5. Sample Size and Randomization

The intervention was done to study the effects of replacing animal-based proteins with plant-based proteins on food and nutrient intake, but also on risk factors of colorectal cancer, type 2 diabetes, and cardiovascular diseases. Sample size was determined to show the effect of dietary intervention on the concentration of haem-derived N-nitroso compounds in faeces, since it was expected to be the variable with the largest variation. The power calculation was based on the data of our previous 4-week intervention study, where the concentration of haem-derived N-nitroso compounds was 2.875 ± 2.800 pmol/mg faeces before the intervention period and 2.245 ± 1.125 pmol/mg faeces at the end of the intervention. The power calculation showed that *n* = 50/group would show the statistical difference between the intervention groups at the end of the intervention period at 95% confidence interval and statistical power of 0.80.

The intervention was carried out staggered so that participants started their 12-week intervention period during 11 consecutive weeks between January and March 2017. Equal number of participants in each diet group started the intervention period each week. Screening was continued until the beginning of March 2017 and participants who proved to be eligible were randomised to the diet groups. The principal investigator generated the allocation sequence and randomly assigned the participants to the intervention groups within a similar deviation of age and gender.

Blinding was not possible due to the nature of the intervention. However, the participants were not aware of the exact proportion of animal to plant-based proteins in their diet because only colour codes were used to mark the diets.

### 2.6. Statistical Analyses

One-way analysis of variance (ANOVA) was used to analyse differences between the intervention groups in all continuous variables at the baseline, as well as differences in nutrient intakes both at the baseline and at the end. Data on education was dichotomised to basic education or secondary/university level education, and differences between the groups in gender and education were analysed using Chi-square test. Differences in plasma lipoproteins between the groups at the end of the intervention were assessed using analysis of covariance (ANCOVA), where the baseline concentration of each lipoprotein was used as a covariate. Age, gender, BMI and use of hormonal medication were also assessed as covariates but did not change the significance of the results. Variables that were not normally distributed were analysed using logarithmic (log10) modifications; all nutrient intakes were log-transformed. Spearman correlation was used to analyse correlations between protein intake calculated based on food record data and 24-h urinary nitrogen excretion. Bonferroni test was used to carry out *post hoc* comparisons between the groups, and in all tests, differences were considered significant at *p* < 0.05. All statistical analyses were performed using SPSS Statistics version 25 (IBM, Armonk, NY, USA).

## 3. Results

### 3.1. Baseline Characteristics

Altogether 136 participants (107 women, 29 men) completed the 12-week intervention period, whereas nine participants (6.2%) dropped out of the study as shown in the flow chart (Figure 1).

The mean age of the participants was 48 years (range 20–69). Most of them had bachelor’s or master’s degree (57%) and were employed (60%) or retired (19%). Based on the data collected using the background questionnaire, they had moderate-intensity physical activity on average three times per week. They were asked to keep their physical activity at the same level as previously throughout the study. There were no differences between the intervention groups in any of the baseline characteristics shown in Table 2.

### 3.2. Protein and Energy Intake

At the baseline, there were no differences in energy and nutrient intakes between the groups (Table 3). Energy intake did not differ between the diet groups at the endpoint of the study (*p* > 0.05; Appendix A), and there were no differences between the groups in the BMI at the endpoint either (ANIMAL 25.0 ± 4.2, 50/50 24.5 ± 4.1, and PLANT 25.2 ± 3.7 kg/m^2^; *p* > 0.05). Based on the food records at the end, the intended protein intake of 17 E% was achieved in the ANIMAL (18.2 ± 3.1 E%, mean ± SD) and 50/50 (16.9 ± 2.2 E%) groups, when the mean intakes are examined. In the PLANT diet group, the mean protein intake was lower (15.2 ± 2.0 E%) in comparison to the ANIMAL and 50/50 diet groups (*p* ˂ 0.001 and *p* = 0.002, respectively; Figure 2A). Protein intake based on food record data was confirmed using urinary nitrogen excretion, a physiological biomarker of protein intake. Based on food records vs. urinary nitrogen excretion, protein intake was 99.2 ± 28.9 vs. 99.3 ± 26.2 g/d in the ANIMAL, 85.5 ± 18.0 vs. 86.5 ± 22.6 g/d in the 50/50, and 80.9 ± 15.1 vs. 77.5 ± 12.4 g/d in the PLANT group. Based on both analysing methods, protein intake in grams was higher in the ANIMAL diet in comparison to the 50/50 and PLANT diets, whereas no difference between the 50/50 and PLANT diets was observed (*p* values for food record data and urinary nitrogen excretion: ANIMAL vs. 50/50 *p* = 0.013 and *p* = 0.014; ANIMAL vs. PLANT *p* < 0.001 for both; 50/50 vs. PLANT *p* = 0.858 and *p* = 0.195). Protein intakes analysed based on food records and calculated from urinary nitrogen excretion correlated significantly in all diet groups (ANIMAL *r* = 0.368, *p* = 0.012; 50/50 *r* = 0.466, *p* = 0.001; PLANT *r* = 0.416, *p* = 0.005).

At the baseline, the proportion of dietary animal proteins among the study participants was on average 62.4% and that of plant proteins 37.6% of total protein intake. The ANIMAL diet was planned to contain 70% of animal-based proteins and 30% of plant-based proteins, and the obtained proportions were 68% and 32%. Respectively, in the 50/50 diet the obtained proportions were 52% of animal and 48% of plant-based proteins. The PLANT diet, intended to contain 30% of animal and 70% of plant-based proteins, provided 26.4% of proteins from animal-based sources and 73.6% from plant-based sources. The food groups sugar and confectionery, fat spreads, oils and dressings, and miscellaneous (see Table 4) were not included in either animal or plant-based protein sources in the calculations.

Dietary protein sources in the intervention diets are shown in Table 4. In the ANIMAL diet, meat dishes together with milk and dairy products provided nearly 60% of total protein intake on average, while their combined share in the 50/50 diet was less than 40%, and in the PLANT diet 15.9% only. The proportions of vegetables and vegetable dishes, nuts and seeds, but also of cereals and bakery products, increased when animal-based protein sources were replaced with plant-based ones. The standardization of the amount of fish and eggs in the diets can be observed as the nearly equal proportions to total protein intake in each intervention group. Appendix A shows that cereals and bakery products were the most important source of energy in all diets. Meat dishes were among the important sources of energy in the ANIMAL and 50/50 diets whereas vegetables and vegetable dishes were among the important ones in the PLANT and 50/50 diets. Milk and dairy products provided a reasonable amount of energy in the ANIMAL diet while nuts and seeds contributed to the energy intakes in the PLANT diet.

### 3.3. Intake of Total Fat, Fatty Acids and Cholesterol

Replacing animal-based protein sources with plant-based ones changed the fatty acid composition of the diets (Figure 2C,E,F) but had no effect on total fat intake (*p* > 0.05). The intake of SFA (E%) was lower (Figure 2C) and the intake of PUFA (E%) higher (Figure 2E) in the PLANT (*p* < 0.001, *p* < 0.001) and 50/50 diet groups (*p* < 0.001, *p* < 0.001) in comparison to the ANIMAL group, but also in the PLANT group when compared with the 50/50 group (*p* = 0.014, *p* < 0.001). The intakes of n-3 (Figure 2F) and n-6 PUFA as well as alpha-linolenic acid were also higher in the PLANT (*p* < 0.001, *p* < 0.001, *p* < 0.001) and 50/50 (*p* = 0.002, *p* < 0.001, *p* = 0.001) groups when compared with the ANIMAL group. Cholesterol intake (mg/d) was lower both in the PLANT (*p* < 0.001) and in the 50/50 groups (*p* < 0.001) than in the ANIMAL group, and lower in the PLANT than in the 50/50 group (*p* < 0.001).

The sources of total fat and fatty acids are shown in Appendix A. In the ANIMAL group, meat dishes were an important source of total fat, SFA, PUFA, and n-3 PUFA. In the PLANT group, nuts and seeds contributed highly to total fat and PUFA as well as to n-3 PUFA intakes. In the PLANT and 50/50 groups, vegetables and vegetable dishes provided a high proportion of total fat and therefore contributed to the intakes of SFA, PUFA, and n-3 PUFA. Dairy products were among the most important sources of saturated fat, and fish dishes were among the most important sources of n-3 PUFA in all diet groups. Cereals and bakery products as well as fat spreads, oils and dressings were a significant source of fat in all diet groups and thus provided SFA, PUFA, and n-3 PUFA.

### 3.4. Intake of Carbohydrates and Fibre

There were no differences in the total intake of carbohydrates between the groups at the endpoint of the intervention (*p* > 0.05; Appendix A). However, despite an average fibre intake of 28.7 g/day among study participants at the baseline, replacing animal-based proteins with plant-based ones increased the fibre intake both in the PLANT (*p* < 0.001) and in the 50/50 groups (*p* = 0.012) when compared to the ANIMAL diet group (Figure 2B). Appendix A shows the proportional sources of carbohydrates and fibre in the intervention diets. The most important sources of carbohydrates and fibre in all diet groups were cereals. Fruits and berries also provided a reasonable amount of those nutrients in all groups. Vegetables and vegetable dishes were a source of fibre in all groups, and in the 50/50 and PLANT groups also important sources of carbohydrates whereas in the ANIMAL group also meat dishes and milk and dairy products contributed highly to the share of carbohydrates.

### 3.5. Plasma Lipoprotein Profile

At the baseline, there were no differences in the plasma lipoprotein profiles between the groups (*p* > 0.05). Replacing animal-based protein sources with plant-based ones decreased the concentrations of total and LDL cholesterol in plasma (*p* = 0.003 between the ANIMAL and PLANT diets for both markers) but had no effect on concentrations of HDL cholesterol or triglycerides (*p* > 0.05) (Figure 3).

## 4. Discussion

In this 12-week clinical trial, we investigated the effects of replacing animal-based proteins with plant-based protein sources on intakes of energy-yielding nutrients, fibre, and plasma lipoproteins among healthy Finnish adults. When designing the study diets, we used a whole-diet approach to be able to control all dietary protein sources and thus to give an overall view of the effects of replacing animal proteins with plant protein sources in a typical Nordic diet. We found that replacing dietary animal proteins with plant protein sources slightly reduces protein intakes, increases fibre intake, improves dietary fatty acid composition and leads to a more favourable plasma lipoprotein profile.

The intervention was implemented using a whole-diet approach, in which red and white meat, as well as dairy products, were partly replaced with a variety of plant-based foods, including legumes, nuts, seeds, soy, and cereals. The amount of fish and eggs were kept the same in all intervention diets. We aimed to cover 30%, 50%, and 70% of total protein intakes with plant-based proteins in the diets, and achieved proportions of 32%, 52%, and 74% of plant-based proteins, and 68%, 48%, and 26% of animal-based proteins, respectively. This reflects the excellent compliance of study participants with their assigned diets, which was also seen as an exceptionally small number (6.2%) of dropouts during the 12-week intervention period. Furthermore, concordant results of protein intake analysed using both food records and 24-h urinary nitrogen excretion—a biomarker of protein intake—also indicated good compliance of the participants. The intervention also produced valuable real-life experiences on feasibility of plant-protein based diets in comparison to an average—predominantly animal-protein based—Finnish diet. The diets containing more plant-based foods were feasible and acceptable among healthy Finnish adults.

Even though the participants were very committed to following their diets, some of them reported problems related to the consumption of certain plant-based products, most often because of intestinal discomfort. This may at least partly explain the lower protein intake in the intervention groups with higher proportion of plant-based proteins, even though the protein intakes were still close to the intended level (15 E% in the PLANT and 17 E% in the 50/50 groups in comparison to 18 E% in the ANIMAL group), and within the recommended range of 10–20 E% for adults. These results are in line with the findings of EPIC-Oxford cohort study, where protein intakes were lower among vegans and vegetarians than among meat-eaters [21]. However, for planning purposes, protein intake of 15 E% is considered a suitable target for adults and 18 E% for individuals over 65 years [22]. Therefore, achieving desired protein intake from diets high in plant-based proteins might be challenging among older people. Adequate protein intake from plant-based diets is also important because of lower digestibility and bioavailability of plant proteins in comparison to animal proteins [23]. In addition, with the exception of soy, providing all essential amino acids with plant-based proteins requires sufficient diversity of plant protein sources [24]. Intestinal discomfort and symptoms may limit the use of plant-protein based foods particularly in populations, where the use of e.g., legumes on typical diets is low.

When animal-based protein sources were replaced with the plant-based ones, the role of meat and dairy products decreased and the role of vegetable dishes including legumes increased as a source of protein. The role of cereals remained seemingly stable. These changes in the diet composition also caused differences in the nutritional quality of the diets. The quality of fatty acid composition of the diet improved, as indicated by a decrease in intake of saturated fatty acids and an increase in intake of polyunsaturated fatty acids. Also, Sobiecki et al. [21] observed in the EPIC-Oxford cohort that vegans had the highest intakes of polyunsaturated fatty acids whereas meat eaters had the highest intake of saturated fatty acids. Since our participants were asked to keep the use of fat on bread and cooking unchanged, the changes in dietary fatty acid composition were due to the increased consumption of vegetable dishes and nuts and seeds, and the decreased consumption of meat and meat products, as well as dairy products, particularly cheese. When replacing red meat with plant-based protein sources, the type of meat may also affect the change in dietary fatty acid composition: replacing regular fat meat with plant-based protein sources may induce larger change in dietary fatty acid composition than replacing lean meat with plant-based products. Since egg consumption was equal in the intervention diets, the dietary changes mentioned above were also a consequence of lower cholesterol intake in the diets containing more plant-based protein sources.

The changes in dietary fatty acid composition presumably resulted in the decrease in both total and LDL cholesterol in plasma. The decrease in the cholesterol intake may also have had a role in these changes at least in some individuals. Our result is in line with earlier meta-analysis data on RCT’s related to the effects of plant-based diets on lipoprotein profile [14,15] as well as what has been observed in vegans [25]. Overall, our results support epidemiological findings on a decreased risk of CVD associated with plant protein intake e.g., [26], which thus can be explained by the changes in the composition of whole diet and not by protein *per se*.

Plant protein sources are typically rich in dietary fibre. Increased proportion of plant protein sources in the 50/50 and PLANT diets led to the concomitant increase in intake of dietary fibre in these groups when compared to the ANIMAL diet group. The significant increase in fibre intake was seen even though the fibre intake of study participants at the baseline (29 ± 10 g/d) was higher than in Finnish adults on average (20 g/d in women, 22 g/d in men) [3] and was already within the recommended range of 25–35 g/d. Our results are in line with those obtained in the EPIC-Oxford cohort where meat eaters had lower fibre intake than vegetarians or vegans [21]. According to the latest report of the World Cancer Research Fund’s expert panel [27], there is strong evidence that consumption of wholegrains and foods containing dietary fibre probably decreases and consumption of red meat probably increases the risk of CRC. In their substitution model, Liao et al. (2019) found a reduction in CRC risk associated with the substitution of protein from bread, cereals and pasta for protein from red meat, which was probably due to both increased fibre intake and decreased intake of meat components, including haem. Similar changes in nutrient intakes took place in our study, indicating that even partial replacement of animal proteins with plant protein sources may potentially reduce CRC risk.

The Planetary Health Diet presented by the EAT-Lancet Commission is a theoretical reference for a healthy and sustainable diet [1], but its nutritional safety and adequacy has not been proved in practice yet. The comparisons presented below are based on estimations, but the PLANT diet in the current study, containing 70% of proteins from plant-based sources and 30% from animal-based sources, could be regarded as a Nordic modification of the Planetary Health Diet. The major source of energy in both our PLANT diet and the Planetary Health Diet are wholegrains and cereals, providing 32% of total energy intake. Vegetables, legumes, potatoes, fruits, and berries provided altogether 27% of total energy on average in our PLANT diet, whereas their proportion in the Planetary Health Diet would be 21%. The proportion of nuts and seeds has been suggested to be larger than it was in the current study’s PLANT diet (11.6% vs. 7.6% of total energy). For cultural and environmental reasons, partial replacement of the number of nuts and seeds in the Planetary Health Diet with other plant-based protein sources in Nordic countries would be reasonable, as the local interpretation and adaptation of the Planetary Health Diet is recommended [1].

Planetary Health Diet is a flexitarian diet containing small amounts of animal-based foods [1]. In comparison to the diets in the current study, the proportions of meat and fish in the Planetary Health Diet might be approximately equal to those of meat and fish dishes in the PLANT diet. The amount of fish was standardised to be the same in all our intervention diets and corresponding to the minimum recommended consumption—two meals containing fish per week—in Finnish national nutrition recommendations [4]. In Finland as well as in other Nordic countries the consumption of local, wild fish is highly recommended for both nutritional and sustainability reasons [4], and the fish consumption could be larger than in the current study diets. The proportion of dairy products in the PLANT diet was even smaller than has been suggested in the Planetary Health Diet. Dairy products in the 50/50 diet contributed 8% of the total energy intake, which could be considered as a justified consumption in countries such as Finland, where dairy cattle are fed with grass grown on arable area that is not suitable for other food production. Overall, the trend for the same favourable changes in fibre intake and fat quality as in the PLANT group were also seen in the 50/50 group, suggesting that relatively moderate and easily achievable changes in the diet may have health benefits.

In our study, the amount of red and processed meat was reduced to one third in the PLANT diet compared to the ANIMAL diet, which was, for the most part, complemented by increasing the number of legume-based dishes. Interestingly, the Planetary Health Diet highlights an increased consumption of legumes as one way to move towards sustainable and healthy diets [1]. Based on the latest FinDiet study [3], the consumption of legumes and legume-based products were very low in Finland, contributing only 1% of total energy in both men and women, and being far from the recommended goals. Our results show that a substantial increase in legume consumption and a decrease in red meat consumption is feasible in Nordic dietary setting, paving the way for a much-needed shift towards more plant-based diets [6]. Nevertheless, the transformation at the population level takes time and we need to think appropriate measures to promote the change.

To our knowledge, this is the first clinical trial that has used a whole-diet approach in studying the effects of replacing animal proteins with plant protein sources among healthy volunteers. Strengths of the study are the intervention period long enough to clearly show the effects of diets on nutrient intakes and plasma lipoprotein profile, the high compliance rate with only 6% of dropouts, and the excellent realization of the planned intervention diets. The good compliance rate was due to motivated and well-educated participants able to carefully follow the intervention diets. However, the higher than average education level of the study participants may limit the external validity of the results. Importantly, the study produced novel information on the overall effects of replacing animal-based proteins with variety of plant-based protein sources on nutrient intakes and blood lipid profile among healthy volunteers, and thus provides knowledge that can be utilised in planning population-level actions towards more sustainable diets.

## 5. Conclusions

Replacing animal protein sources with plant-based ones tends to decrease protein intake but still provides the recommended intake for a working-age population. Increasing the proportion of plant protein sources in diet results in an increase of fibre intake and improves the quality of dietary fat by reducing the intake of saturated fatty acids and increasing the intake of polyunsaturated fatty acids. The changes in the quality of dietary fatty acid composition leads to a significant improvement in blood lipoprotein profile. Flexitarian diets—such as intervention diets in the current study—would provide healthy and more sustainable alternatives for current, mostly animal-based diets. In particular, a diet containing one-half of the proteins from animal-based sources and the other from plant-based sources would be easily realisable at the population level. However, adequate intakes of vitamins such as vitamin B_12_ and vitamin D, as well as minerals including iron, iodine, and calcium, have to be confirmed.

## Figures and Tables

**Figure 1 nutrients-12-00943-f001:**
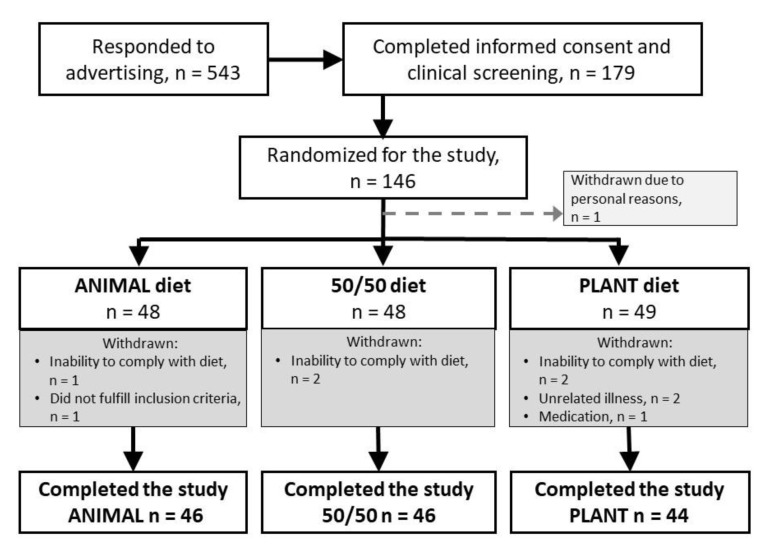
Flow chart of the participants. Dropouts are shown on grey background.

**Figure 2 nutrients-12-00943-f002:**
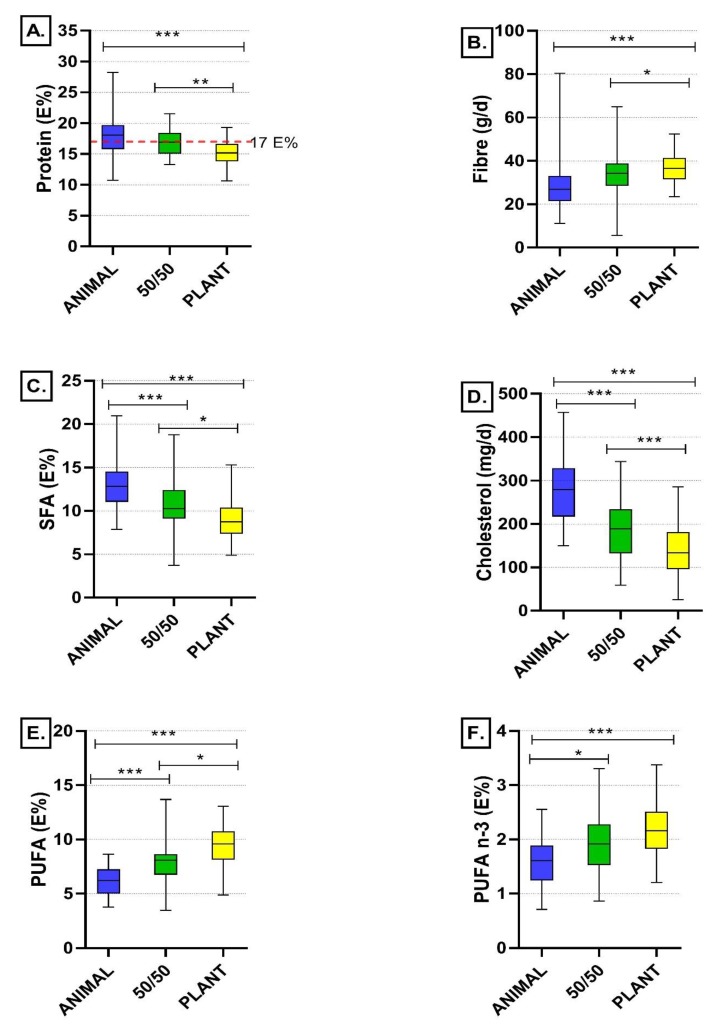
Endpoint intakes of (**A**) protein (E%), (**B**) fibre (g/d), (**C**) saturated fatty acids (SFA) (g/d), (**D**) cholesterol (mg/d), (**E**) polyunsaturated fatty acids (PUFA) (E%), and (**F**) n-3 polyunsaturated fatty acids (PUFA n-3) (E%) in the diet groups. Differences between the diet groups analysed by ANOVA with Bonferroni correction: * *p* < 0.05, ** *p* < 0.01, *** *p* < 0.001.

**Figure 3 nutrients-12-00943-f003:**
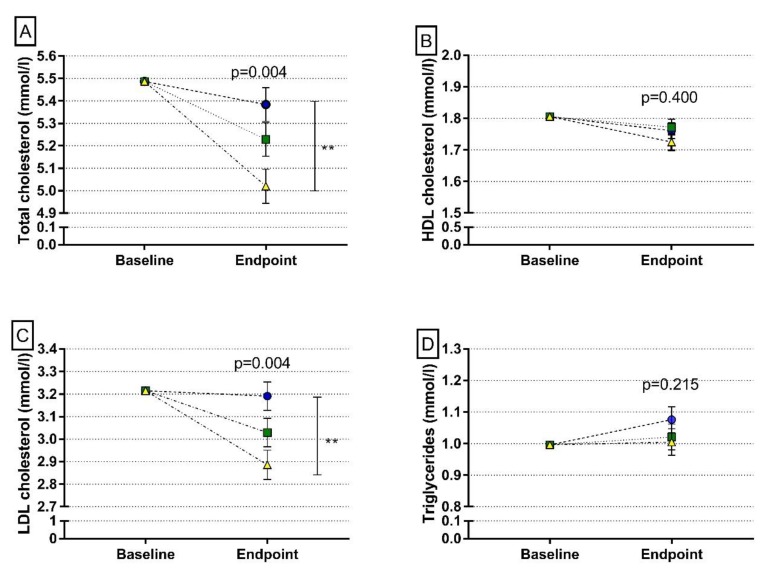
Plasma lipids at the endpoint of the intervention: (**A**) total cholesterol (**B**) high-density lipoprotein (HDL) cholesterol, (**C**) low-density lipoprotein (LDL) cholesterol, (**D**) triglycerides. Symbols: BLUE circle = ANIMAL (a diet containing 70% animal-based and 30% plant-based protein sources of total protein intake); GREEN square = 50/50 (a diet containing 50% animal-based protein sources and 50% plant-based protein sources of total protein intake); YELLOW triangle = PLANT (a diet containing 70% plant-based and 30% animal-based protein sources of total protein intake); ** *p* < 0.01 between a diet containing 70% plant-based protein sources of total protein and a diet containing 70% animal-based protein sources of total protein. *p* values from ANCOVA, adjusted for baseline, Bonferroni correction used for group comparison.

**Table 1 nutrients-12-00943-t001:** Consumption frequencies and daily consumption of specific foods and food groups in the intervention diets based on the delivered food items and diet instructions. Average daily consumption is presented as g/day.

	ANIMAL	50/50	PLANT
Sources of animal proteins Main dishes containing minced meat (times/wk) Main dishes containing whole meat (times/wk) Main dishes containing sausage (times/wk) Sausages and cold cuts, including processed poultry (g/day) Pork and beef (g/day) Red and processed meat total (g/day)	2–3 2–3 1 35 64 99	1–2 1–2 0–1 23 43 65	0–1 0–1 0–1 11 21 32
Fish dishes (times/wk) Fish (g/day)	2 36	2 36	2 36
Main dishes containing poultry (times/wk) Poultry (g/day)	2–3 43	1–2 29	1 14
Eggs/wk (in dishes and pastries; boiled or fried) Eggs (g/day)	4 31	4 31	4 31
Dairy products other than cheese (g/day)	400	250	125
Cheese (g/day)	40	25	10–15
Sources of plant proteins Main dishes based on peas, lentils, chickpeas, tofu, crushed soya beans or faba beans as main ingredients (times/wk)	0–1	3–5	5–7
Vegetable patties, pizza, mushroom dishes (portions/wk)	1	2–3	2–3
Nuts, almonds, and seeds (g/day)	*	16	34
Plant-based dairy-like products (other than cheese), g/day	0	150	250
Bread (rye and oat/wheat bread; slices of bread/day) Bread (g/day)	4–5 120–150	6 180	7 210
Porridge and muesli (g/day, dry weight)	40	40–60	40–80
Wholegrain rice, pasta, couscous and quinoa (g/day, dry weight)	70	70–105	70–140
Potatoes (g/day, cooked)	120	0–120	0–120

*** Occasional consumption of nuts, almonds, and seeds was allowed in ANIMAL group.

**Table 2 nutrients-12-00943-t002:** Participant characteristics at the baseline.

	ANIMAL (*n* = 46)	50/50 (*n* = 46)	PLANT (*n* = 44)
Age (years)	47.6 ± 14.5	47.2 ± 14.7	48.7 ± 14.0
Classified age (n, %)			
20–29	7 (15.5)	7 (15.2)	5 (11.4)
30–39	9 (19.6)	8 (17.4)	7 (15.9)
40–49	8 (17.4)	9 (19.6)	10 (22.7)
50–59	11(23.9)	10 (21.7)	11 (25.0)
60–69	11(23.9)	12 (26.1)	11 (25.0)
Gender (n, %)			
Female	37 (80.4)	36 (78. 3)	34 (77.3)
Male	9 (19.6)	10 (21.7)	10 (22.7)
Education ^1^ (n, %)			
Basic education	10 (21.7)	13 (28.3)	7 (15.9)
Secondary/university level education	33 (71.7)	26 (56.5)	35 (79.5)
BMI (kg/m^2^)	24.6 ± 4.1	24.3 ± 4.0	25.2 ± 3.8
Cholesterol (mmol/L)			
Total	5.4 ± 1.1	5.6 ± 0.8	5.5 ± 1.0
LDL	3.2 ± 0.9	3.3 ± 0.8	3.2 ± 0.9
HDL	1.8 ± 0.4	1.8 ± 0.4	1.9 ± 0.4
Triglycerides (mmol/L)	1.1 ± 0.4	1.0 ± 0.4	1.0 ± 0.3

Values are presented as mean ± SD, except for (classified) age and gender, which are presented as total number and percentage value (n, %). Differences between the groups were analysed using ANOVA for continuous variables and Chi-square test for dichotomous variables (gender and education), *p* > 0.05 for all variables. ^1^ For education data, total *n* = 124 (data is missing from 12 subjects). LDL low-density lipoprotein; HDL high-density lipoprotein.

**Table 3 nutrients-12-00943-t003:** Daily energy and nutrient intake in the intervention groups at the baseline; calculated as an average of 4-day food records. *p* values (ANOVA) between the groups >0.05.

	ANIMAL (*n* = 45)	50/50 (*n* = 46)	PLANT (*n* = 43)
Energy intake (kJ)	9124 ± 2179	8731 ± 1959	8788 ± 1675
Energy intake (kcal)	2180 ± 520	2087 ± 468	2101 ± 400
Protein (E%)	18.5 ± 3.2	17.3 ± 3.1	17.7 ± 3.4
Carbohydrates (E%)	39.0 ± 6.6	40.3 ± 5.8	40.7 ± 5.7
Fat (E%)	37.7 ± 5.7	37.2 ± 6.1	36.6 ± 5.2
Saturated fatty acids (E%)	13.2 ± 2.9	13.1 ± 4.3	12.3 ± 2.8
Monounsaturated fatty acids (E%)	12.9 ± 2.6	12.7 ± 2.7	12.9 ± 2.6
Polyunsaturated fatty acids (PUFA)			
Total PUFA (E%)	6.2 ± 2.0	6.3 ± 1.5	6.6 ± 1.8
n-6 PUFA (E%)	4.6 ± 1.4	4.6 ± 1.2	4.7 ± 1.3
n-3 PUFA (E%)	1.7 ± 0.7	1.6 ± 0.6	1.7 ± 0.6
α-linolenic acid (E%)	1.1 ± 0.5	1.1 ± 0.4	1.1 ± 0.4
Cholesterol (mg)	315 ± 145	295 ± 129	282 ± 127
Fibre (g)	30.0 ± 12.3	28.4 ± 8.7	27.7 ± 8.4

**Table 4 nutrients-12-00943-t004:** Sources of protein in the intervention diets presented as average proportions (% ± SD) based on 4-day food record data.

Food Group	ANIMAL	50/50	PLANT
Meat dishes	34.0 ± 10.5	24.5 ± 11.7	10.3 ± 7.7
Milk and dairy products	23.3 ± 8.2	12.2 ± 6.0	5.6 ± 4.3
Fish dishes	6.1 ± 4.6	6.8 ± 5.7	7.4 ± 5.2
Egg dishes	2.5 ± 2.9	2.4 ± 2.6	1.9 ± 2.2
Cereals and bakery products	22.7 ± 7.1	25.8 ± 8.2	28.5 ± 9.8
Vegetables and vegetable dishes	4.8 ± 3.6	17.9 ± 9.2	28.6 ± 12.5
Nuts and seeds	0.9 ± 1.6	2.1 ± 3.0	7.0 ± 3.9
Plant-based dairy-like products	0.0 ± 0.1	1.7 ± 2.2	3.6 ± 5.0
Potatoes and potato dishes	0.9 ± 1.0	0.7 ± 1.2	0.4 ± 0.7
Fruits, berries, fruit and berry dishes	1.6 ± 1.0	1.9 ± 1.3	2.0 ± 1.2
Beverages	1.9 ± 1.0	2.0 ± 2.2	2.4 ± 1.2
Sugar and confectionery	0.8 ± 0.9	1.2 ± 1.5	0.6 ± 1.0
Miscellaneous ^1^	0.3 ± 0.8	0.4 ± 0.9	1.2 ± 2.9
Fat spreads, oils and dressings	0.2 ± 0.2	0.4 ±0.7	0.5 ± 1.3

^1^ Food group “Miscellaneous” includes dried fruits and berries, snacks, spices, piquant sauces, weight loss products, meal replacements, protein powders, protein bars, and other miscellaneous foods.

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
