# Peer review of "Replacing Animal-Based Proteins with Plant-Based Proteins Changes the Composition of a Whole Nordic Diet—A Randomised Clinical Trial in Healthy Finnish Adults"

_nutrients, 2020, doi:10.3390/nu12040943_

Round 1

Reviewer 1 Report

The aim of the study was to investigate the effects of partial replacement of dietary animal proteins with plant-based ones on intake of energy yielding nutrients, fibre, and plasma lipoproteins.

Introduction provides sufficient background and includes relevant references. The research design is appropriate

I recommend Accept after minor revision related to

I suggest modifying the title by Replacing Animal-Based Proteins with Plant-Based Proteins Changes the Composition of a Whole Nordic Diet—A Randomized Clinical Trial in Healthy Finnish Adults

I suggest including a table similar to Table 2 including the participant characteristics at the endpoint of the study

I suggest including some lines about lifestyle or physical activity of the participants

English language and style are fine spell check required

Reviewer 2 Report

Your manuscript was of the highest quality - each section was was very clear and scientifically sound.  I can not add any comments or recommended changes.  I found the manuscript of high interest.  

1. Figure 3 is unnecessary as the % of animal and plant based protein can be added to Table 3. 2. Figure 4 would be clearer if you use bar rather than line graphing 3. Suggest comments on lean vs regular fat meat vs plant proteins.
